# Analysis of Precise Orbit Determination for the HY2D Satellite Using Onboard GPS/BDS Observations

**Hailong Peng [1,2], Chongchong Zhou [3,4,*], Shiming Zhong [3], Bibo Peng [3,4], Xuhua Zhou [5], Haoming Yan [3], Jie Zhang [3], Jinyang Han [3,6], Fengcheng Guo [7] and Runjing Chen [8]**

1  National Satellite Ocean Application Service, No. 8 Dahuisi Road, Haidian District, Beijing 100081, China; phl@mail.nsoas.org.cn
2  Key Laboratory of Space Ocean Remote Sensing and Application, MNR, No. 8 Dahuisi Road, Haidian District, Beijing 100081, China
3  State Key Laboratory of Geodesy and Earth's Dynamics, Innovation Academy for Precision Measurement Science and Technology, Chinese Academy of Sciences, No. 340 Xudong Road, Wuhan 430077, China; smzhong@apm.ac.cn (S.Z.); pengbibo@apm.ac.cn (B.P.); yhm@apm.ac.cn (H.Y.); zhangjie@apm.ac.cn (J.Z.); goldensun_han@apm.ac.cn (J.H.)
4  National Geodetic Observatory, Wuhan, Innovation Academy for Precision Measurement Science and Technology, Chinese Academy of Sciences, No. 340 Xudong Road, Wuhan 430077, China
5  Shanghai Astronomical Observatory, Chinese Academy of Sciences, No. 80 Nandan Road, Shanghai 200030, China; xhzhou@shao.ac.cn
6  College of Earth and Planetary Sciences, University of Chinese Academy of Sciences, No. 19A Yuquan Road, Shijingshan District, Beijing 100049, China
7  School of Geography, Geomatic and Planning, Jiangsu Normal University, No. 101 Shanghai Road, Tongshan District, Xuzhou 221116, China; fchguo@jsnu.edu.cn
8  School of Computer and Information Engineering, Xiamen Institute of Technology, No. 600 Ligong Road, Jimei District, Xiamen 361024, China; chenrj@xmut.edu.cn
*  Correspondence: zcc@apm.ac.cn; Tel.: +86-133-9723-4970

**Abstract:** High-precision orbits of Low Earth Orbit (LEO) satellites are essential for many scientific applications, such as assessing the change in current global mean sea level, estimating the coefficients of gravity field, and so on. How to determinate the high-precision orbits for LEO satellites has gradually become an important research focus. HY2D is a new altimetry satellite of China, which is equipped with a Global Positioning System (GPS) and the third generations of the BeiDou Global Navigation Satellite System (BDS-3) in order to guarantee the reliability of orbital precision in radar altimetry mission. Therefore, this study adopts one month of spaceborne data to conduct the research of precise orbit determination (POD) for the HY2D satellite. Our analysis results are: (1) The standard deviation of residuals for the HY2D satellite based on spaceborne BDS and GPS data are 9.12 mm and 8.53 mm, respectively, and there are no significant systematic errors in these residuals. (2) The comparison results with Doppler Orbitography and Radio-positioning Integrated by Satellite (DORIS)-derived orbits indicate that the HY2D satellite, using spaceborne BDS and GPS data, can achieve the radial accuracy of 1.4~1.5 cm, and the mean three-dimensional (3D) accuracy are 5.3 cm and 4.3 cm, respectively, which can satisfy high-precision altimetry applications. (3) By means of satellite laser ranging (SLR), the accuracy of Global Navigation Satellite System (GNSS)-derived orbits of HY2D is approximately 3.3 cm, which reflects that the model strategies are reliable.

**Keywords:** high-precision orbits; HY2D; spaceborne BDS and GPS data; radial accuracy; model strategies

## 1. Introduction

With the rapid development of space technologies, more and more Low Earth Orbit Satellites (LEOs) have been successfully used in scientific missions [1–7]. Especially, in order to obtain more detailed information about the marine environment and mapping and carry out research about the change in current global mean sea level, tens of ocean altimetry

satellites have been launched. For example, there have been successful launches of Seasat, GeoSat, Topex/Poseidon, HY2A/B/C, Jason1, Jason2 and Jason3 altimetry satellites [1,8], which provide a large amount of effective and high-precision data for assessing the change in current global mean sea level. As shown by the successful use of Global Positioning System (GPS) in the precise orbit determination (POD) for Topex/Poseidon satellite [8], the spaceborne GPS technique makes it possible to obtain centimeter-level orbit products and, thus, has been widely used for LEOs [9–15]. Gao et al. [10] adopted DORIS (Doppler Orbitography and Radiopositioning Integrated by Satellite) data of HY2A to analyze POD results, and showed that the radial orbit difference with the CNES (Centre National d'Etudes Spatiales) orbits is about 1.1 cm. Guo et al. [11] conducted POD of HY2A based on GPS and DORIS data, and achieved radial accuracy better than 1.0 cm. Wang et al. [16] adopted three months of GPS observations of HY2C, and achieved the radial accuracy of about 1.2 cm.

China has developed and operated the BeiDou Navigation Satellite System (BDS) independently [17]. Nowadays, there are a total of 34 BDS satellites in orbit, including 15 BDS-2 satellites (six geostationary Earth orbit (GEO) satellites, six inclined geosynchronous orbit (IGSO) satellites, and 3 medium-Earth orbit (MEO) satellites) and 19 BDS-3 satellites (two IGSO satellites and 17 MEO satellites). There are more and more LEO satellites carrying BDS receivers, and numerous studies have been carried out on the POD based on spaceborne BDS data in recent years. In 2013, the FengYun-3C (FY-3C) satellite was launched successfully, which was equipped with BDS-2 and GPS receivers simultaneously. The spaceborne BDS and GPS data of FY-3C satellite provided a great opportunity to analyze the POD performance of LEOs with BDS. Li et al. [18] conducted POD of FY-3C based on BDS-only and BDS/GPS combined; the analysis showed that the BDS-only orbits can reach a three-dimensional (3D) root mean square (RMS) of 8 cm based on the orbit overlap comparison, while the 3D RMS value of combined POD is 3.9 cm. Xiong et al. [19] achieved real-time POD with a precision of 1.24 m for the FY-3C satellite using BDS and GPS pseudo-range observations. Zhao et al. [20] used the derived POD orbits of FY-3C and regional station observations to enhance the BDS orbits and improved the accuracy from 354.3 to 63.1 cm for GEO, 22.70 to 20.0 cm for IGSO, and 20.9 to 16.7 cm for MEO. Based on the FY-3C spaceborne BDS and GPS data from 2013 to 2017, Li et al. [21] found that the combined POD (without GEO) can achieve slightly better precision than the GPS results, which indicates that when high-quality BDS orbit and clock products are used, the combined solution can improve the accuracy of POD for LEOs in comparison with the GPS-only solution. Based on the spaceborne BDS data of FY-3C, Zhang et al. [22] found that the precision of LEO orbit determination and reliability of the solution are improved through the calibration of daily orbit biases in GEO. The measurements of Tianping-1B, launched in 2018, were also collected by Zhao et al. from GPS/BDS-3. Their results indicated that the orbit consistency of the combined BDS-3/GPS solutions was below 3.5 cm [23].

In 2021, China successfully launched a new altimetry, satellite-HaiYang-2D (HY2D), which is a marine operational satellite that can provide precise ocean dynamic environmental information for the warning and forecasting of marine disaster, continuously measuring the sea surface height and sea surface wind and conducting marine scientific research. In order to guarantee the reliability of satellite orbits for radar altimetry mission, HY2D is equipped with a GPS and BDS receiver and carries a laser retro-reflector array for satellite laser ranging (SLR). Therefore, this paper adopts spaceborne GPS and BDS data to conduct research about the POD of HY2D, mainly including the model strategies used in POD processing, Global Navigation Satellite System (GNSS)-derived orbit analysis, and the validation of SLR residuals for HY2D. The relevant research results can lay an important foundation for the development of subsequent altimetry satellites, and that in turn can lay an important foundation for the development of a spaceborne BDS receiver.

This paper is structured as follows. Section 2 introduces materials and mainly consists of general information about the HY2D satellite and data collection. Section 3 mainly presents the POD method and strategies. All results and discussion obtained based on

the above methods and strategies are given in Section 4. Finally, conclusions are given in Section 5.

## 2. Materials

### 2.1. HY2D Spacecraft

On 19 May 2021, the HY2D satellite was launched at the Jiuquan satellite launch center successfully. The HY2D satellite adopts a non-sun-synchronous orbit with the inclination of 66°, and the orbit altitude of about 960 km. The primary sensors comprise radar altimeter, microwave scatterometer, calibration radiometer, data collection system and ship automatic identification system [24].

Figure 1 shows the HY2D spacecraft and its payloads. It should be pointed out that the GNSS antennas consists of GPS and BDS antennas but cannot receive GPS/BDS signals simultaneously. Thus, the GPS and BDS receivers would switch to each other as needed. The X, Y and Z are the three axes of the satellite flight reference frame (SFF) in the figure, and the +X and +Z axes point toward the direction of flight and nadir, respectively. The +Y axis completes the right-hand orthogonal reference. When the satellite is at zero attitude, the SFF coincides with the satellite body reference frame (SBF), while the SFF is different from the SBF in a non-zero attitude. Table 1 lists the coordinates of the GPS and BDS antenna phase center, laser retro-array (LRA) spherical center, and center of mass in the SBF. With the help of these coordinates, we can perform phase center correction of receivers for HY2D in POD processing.

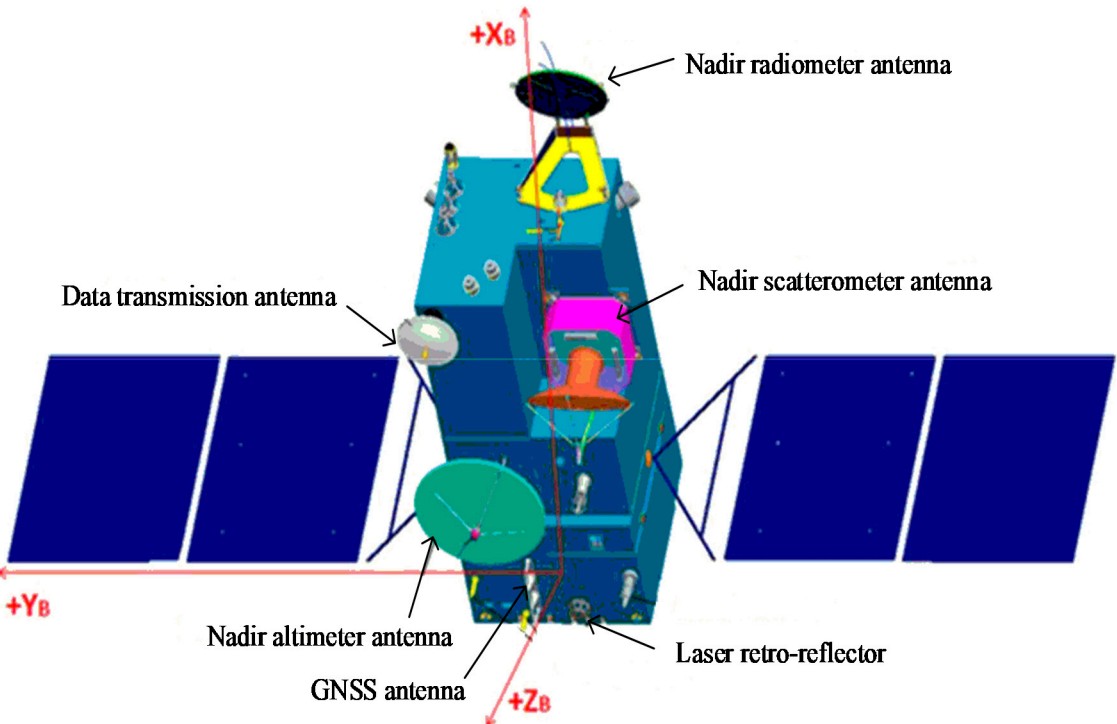

**Figure 1.** HY2D satellite and its payloads.

**Table 1.** Coordinates of the center of mass, GPS and BDS antenna phase centers and LRA spherical center in the SBF.

|  | X (mm) | Y (mm) | Z (mm) |
|---|---|---|---|
| Center of mass | 1319.4 | −4.7 | 5.7 |
| GPS phase center (L1) | 347.2 | −181.9 | −1377.5 |
| GPS phase center (L2) | 347.4 | −181.0 | −1396.1 |
| BDS phase center (B1) | 427.4 | 177.7 | −1378.5 |
| BDS phase center (B2) | 427.7 | 177.9 | −1397.3 |
| LRA spherical center | 311.7 | −215.5 | 1060.8 |

When the satellite is flying, surface forces acting on HY2D are mainly from atmospheric drag, earth radiation pressure and solar radiation pressure [10]. In order to model the forces precisely, having a well knowledge of the characteristics for the HY2D satellite surfaces and radiators is very necessary. Table 2 gives the parameters information of spacecraft surfaces and radiators, including the optical characteristics and projected areas. With help of the information, the solar radiation pressure of HY2D can be described with the Box-Wing model.

**Table 2.** Projected areas and optical characteristics of HY2D.

|  |  |  | X+ | X− | Y+ | Y− | Z+ | Z− | SA+ | SA− |
|---|---|---|---|---|---|---|---|---|---|---|
| Spacecraft surfaces | | Projected area (m$^2$) | 3.62 | 3.92 | 5.17 | 5.46 | 3.06 | 6.22 | - | - |
| | Visible | Specular | 0.65 | 0.65 | 0.65 | 0.65 | 0.65 | 0.65 | - | - |
| | | Diffuse | 0.00 | 0.00 | 0.00 | 0.00 | 0.00 | 0.00 | | |
| | | Absorbed | 0.35 | 0.35 | 0.35 | 0.35 | 0.35 | 0.35 | - | - |
| | Infra-red | Specular | 0.00 | 0.00 | 0.00 | 0.00 | 0.00 | 0.00 | - | - |
| | | Diffuse | 0.31 | 0.31 | 0.31 | 0.31 | 0.31 | 0.31 | | |
| | | Absorbed | 0.69 | 0.69 | 0.69 | 0.69 | 0.69 | 0.69 | - | - |
| Radiators and solar arrays | | Projected area (m$^2$) | 0.33 | 0.37 | 2.61 | 2.33 | 4.88 | 1.72 | 18.12 | 18.12 |
| | Visible | Specular | 0.87 | 0.87 | 0.87 | 0.87 | 0.00 | 0.87 | 0.10 | 0.00 |
| | | Diffuse | 0.00 | 0.00 | 0.00 | 0.00 | 0.15 | 0.00 | 0.00 | 0.10 |
| | | Absorbed | 0.13 | 0.13 | 0.13 | 0.13 | 0.85 | 0.13 | 0.90 | 0.90 |
| | Infra-red | Specular | 0.22 | 0.22 | 0.22 | 0.22 | 0.00 | 0.22 | 0.08 | 0.00 |
| | | Diffuse | 0.00 | 0.00 | 0.00 | 0.00 | 0.15 | 0.00 | 0.00 | 0.10 |
| | | Absorbed | 0.78 | 0.78 | 0.78 | 0.78 | 0.85 | 0.78 | 0.92 | 0.90 |

Table 2 shows that the +X and +Z axes point to the directions of flight and nadir, respectively, and -Y axis points to the direction of Sun. As the HY2D satellite belongs to a non-sun-synchronous orbit satellite, the solar arrays are always directed to the sun. Moreover, the HY2D satellite is in a non-zero attitude during the flight, and the SBF has to rotate according to the attitude following the order of yaw-roll-pitch to obtain the SFF. So, it is essential to adopt satellite attitude data to calculate the solar radiation pressure and phase center offset accurately.

Table 3 shows a part of attitude data of HY2D satellite, which is provided by National Satellite Ocean Application Service (NSOAS) at present. As can be shown from Table 3, the attitude data consist of date, time, roll angle, pitch angle, and yaw angle. In addition, the change in yaw angle works well with the increase in time, so, this study needs to use the attitude data of HY2D to correct the phase center offset and solar radiation pressure.

**Table 3.** The attitude data of HY2D satellite provided by NSOAS.

| Date (Year/Month/Day) | Time (Hour/Minute/Second) | Roll (deg.) | Pitch (deg.) | Yaw (deg.) |
|---|---|---|---|---|
| 2021/7/10 | 19:20:56.322 | −0.0770 | −0.0385 | +88.0330 |
| 2021/7/10 | 19:20:57.346 | −0.0770 | −0.0385 | +88.1155 |
| 2021/7/10 | 19:20:58.370 | −0.0770 | −0.0385 | +88.1980 |
| 2021/7/10 | 19:20:59.394 | −0.0770 | −0.0385 | +88.2805 |
| 2021/7/10 | 19:21:00.418 | −0.0770 | −0.0385 | +88.3630 |
| 2021/7/10 | 19:21:01.442 | −0.0770 | −0.0385 | +88.4455 |
| 2021/7/10 | 19:21:02.466 | −0.0825 | −0.0385 | +88.5225 |
| 2021/7/10 | 19:21:03.490 | −0.0825 | −0.0385 | +88.6050 |
| 2021/7/10 | 19:21:04.514 | −0.0825 | −0.0385 | +88.6875 |
| 2021/7/10 | 19:21:05.538 | −0.0825 | −0.0385 | +88.7700 |
| . . . | . . . | . . . | . . . | . . . |
| 2021/7/10 | 19:30:47.425 | −0.1650 | +0.0165 | +133.0285 |
| 2021/7/10 | 19:30:48.449 | −0.1650 | +0.0165 | +133.1000 |
| 2021/7/10 | 19:30:49.473 | −0.1650 | +0.0165 | +133.1715 |
| 2021/7/10 | 19:30:50.497 | −0.1650 | +0.0165 | +133.2375 |
| 2021/7/10 | 19:30:51.521 | −0.1650 | +0.0165 | +133.3090 |
| 2021/7/10 | 19:30:52.545 | −0.1650 | +0.0165 | +133.3750 |
| 2021/7/10 | 19:30:53.313 | −0.1650 | +0.0165 | +133.4300 |
| 2021/7/10 | 19:30:54.337 | −0.1650 | +0.0165 | +133.4960 |

### 2.2. Data Collection

For evaluating the POD performances of HY2D, we selected spaceborne BDS data from July 6 to July 19, and GPS data from July 20 to August 10 to conduct experiments. Here, Figures 2 and 3 present the number of BDS and GPS satellites observed on a day, respectively.

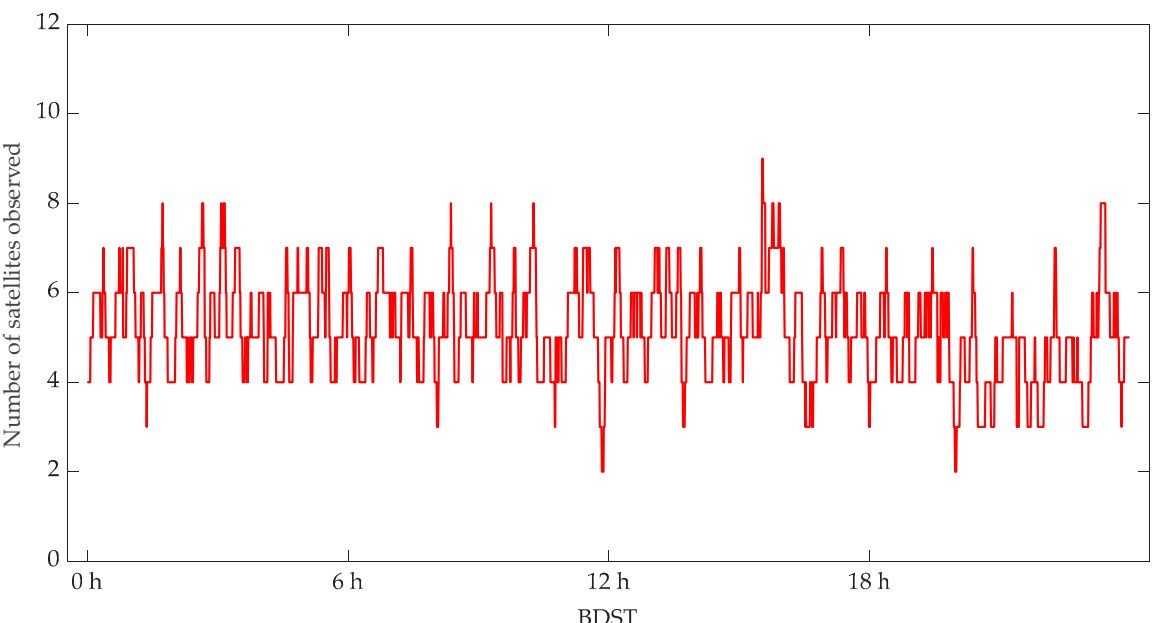

**Figure 2.** Number of BDS satellites observed on 10 July 2021.

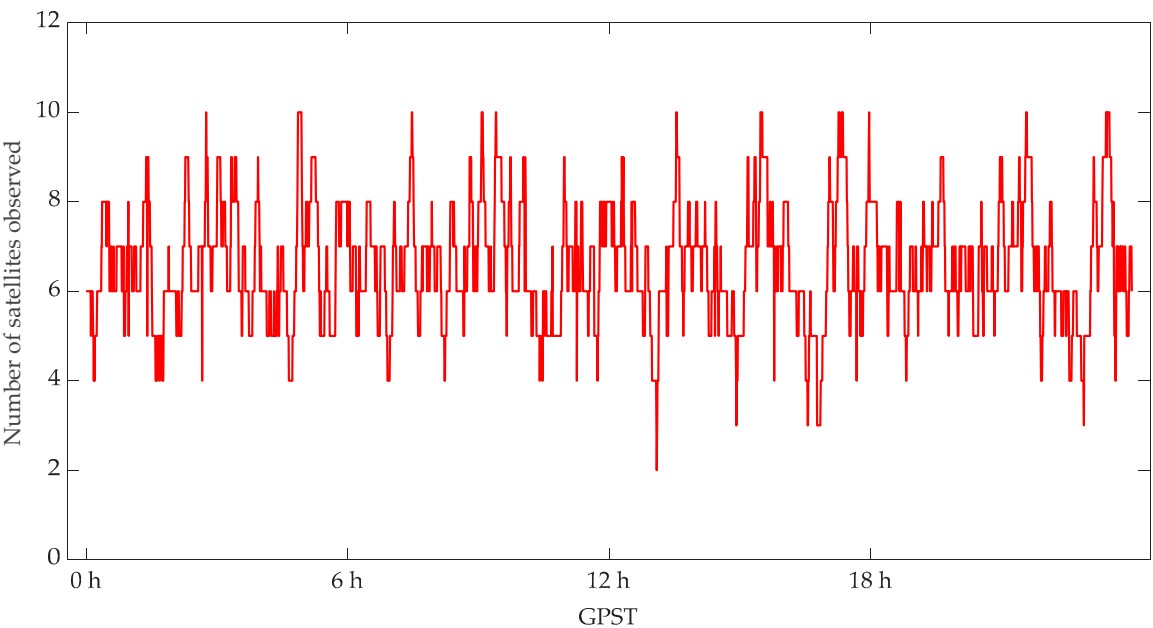

**Figure 3.** Number of GPS satellites observed on 21 July 2021.

As can be shown from Figure 2, most of the epochs have at least four BDS satellites available, and there are an average of five satellites. However, the smallest is 2, and the biggest is 9. As can be seen from Figure 3, compared to BDS, the usability of satellites for GPS is much better, and most epochs have more than five GPS satellites, and the maximum is up to 10 GPS satellites. This is mainly attributed to the GPS constellation consisting of 32 MEO satellites, more than the current BDS constellation, and the fact that most BDS, GEO and IGSO satellites are located within the Asia-pacific region.

## 3. POD Method and Strategies

This Section mainly describes the observation and dynamic model used in the POD processing of HY2D. Meanwhile, we also present the detailed dynamic and observation model in the POD for HY2D.

### 3.1. Observation Model

The GNSS antenna of HY2D satellite only tracks the GPS or BDS signal separately at present, so we selected ionosphere-free combined observation from GPS or BDS to calculate precise orbits. Equation (1) gives ionosphere-free combined observation of GPS and BDS.

$$\begin{cases} P_B = \rho_B - c \cdot \delta t_B^s + c \cdot \delta t_B^t + \varepsilon_{BP} \\ L_B = \rho_B - c \cdot \delta t_B^s + c \cdot \delta t_B^t + N_B \cdot \lambda_B + \varepsilon_{BL} \\ P_G = \rho_G - c \cdot \delta t_G^s + c \cdot \delta t_G^t + \varepsilon_{GP} \\ L_G = \rho_G - c \cdot \delta t_G^s + c \cdot \delta t_G^t + N_G \cdot \lambda_G + \varepsilon_{GL} \end{cases} \tag{1}$$

where $P_B$ and $P_G$ are the code observations of BDS and GPS, respectively, while $L_B$ and $L_G$ are the carrier phase observations of BDS and GPS, respectively; $\rho_B$ and $\rho_G$ are geometrical distance from BDS and GPS signal to receiver, respectively, $\delta t_B^s$ and $\delta t_G^s$ stand for satellite clock errors for BDS and GPS, $\delta t_B^t$ and $\delta t_G^t$ are clock errors for BDS and GPS receivers, $N_B$ and $N_G$ refer to combined ambiguity parameters of BDS and GPS observations separately, $\lambda_B$ and $\lambda_G$ refer to combined wave length of BDS and GPS observations separately, $\varepsilon_{BP}$, $\varepsilon_{BL}$, $\varepsilon_{GP}$ and $\varepsilon_{GL}$ stand for different type observation noise.

### 3.2. Dynamic Model

The equation of motion of a single LEO satellite in the inertial frame can be expressed as follows [25]:

$$\ddot{\vec{r}} = -\frac{GM}{\left|\vec{r}\right|^3}\vec{r} + a_r + a_{rtn} \tag{2}$$

where, *GM* stands for geocentric gravitational constant, $\vec{r}$ and $\ddot{\vec{r}}$ are the position and acceleration of the satellite in the inertial coordinate system, respectively; $a_r$ refers to main perturbation acceleration excluded Earth center gravity, that includes the Sun and Moon perturbation, atmosphere drag, solar radiation pressure, Earth radiation pressure, solid Earth tide, ocean tide and so on. $a_{rtn}$ refers to the periodic radial, tangential, and normal (RTN) perturbation, which can make up for these unmodeled perturbation acceleration errors [26–28]. In this study, the tangential and normal perturbation parameters are estimated for the POD processing.

In order not to affect the calculation efficiency and accuracy of orbit determination, the process of choosing the dynamic and observation model is important. For this purpose, the used dynamic model and measurement model are designed in the POD processing for the HY2D satellite refers to the above descriptions. Furthermore, this study also shows the used model information of the SLR validation for the calculated HY2D orbit. The details are given in Table 4. We adopted the 24 h arc solution based on the dynamical method orbit determination, the atmospheric drag coefficient, the solar radiation pressure coefficient, and RTN perturbation parameters are estimated per 6 h, 24 h, and 24 h, respectively. Moreover, the receiver clock parameter for HY2D is estimated with the gaussian white noise model, and the float solutions of ambiguities are also estimated.

**Table 4.** Dynamical and observation model employed in the POD for HY2D.

| Project | Selection/Description |
|---|---|
| **Dynamic model** | |
| Gravity model | EIGEN-GRGS.RL04. MEAN-FIELD 120 × 120 [29] |
| Atmosphere drag | MSIS00 density model [30] |
| Solar radiation pressure | Box-Wing model [31] |
| Sun and moon ephemeris | JPL DE405 [32] |
| Earth radiation pressure | Knocke-Ries-Tapley model [33] |
| Empirical force model | RTN perturbation |
| Ocean tide [34] | FES2004 [35] |
| Solid Earth tide [34] | TIDE2000 [34] |
| Earth orientation parameter | IERS EOP 14 C04 [36], IAU2000A model |
| **Observation model** | |
| Data type | Code and phase observation of ionosphere-free combination |
| Data interval | 30 s |
| Elevation cutoff | 7° |
| HY2D satellite attitude | Quaternion data |
| GPS and BDS phase model | IGS14.atx |
| Orbit determination arc length and integration step | 24 h arc dynamic solution, 30 s integral step |
| GPS and BDS satellite ephemeris and clock | CODE precise products |
| Atmospheric drag coefficient estimation | Cd/6 h |
| Solar radiation pressure coefficient estimation | Cr/24 h |
| RTN perturbation estimation | Tangential and normal/24 h |
| Receiver clock error estimation | Gaussian white noise |
| Ambiguities | Float solutions of ionosphere-free combination |

**Table 4.** *Cont.*

| Project | Selection/Description |
|---|---|
| SLR validation for the calculated HY2D orbit | |
| Cut-off angle | 20° |
| SLR station coordinate | SLRF2014 |
| Troposphere delay correction | Mendes-Pavlis delay model [37] |
| Relativity correction (propagation path) | IERS Conventions 2010 [34] |

## 4. Results and Discussion

For the sake of analyzing the orbit determination accuracy of the HY2D satellite, we selected spaceborne BDS data from 6 July to 19 July, and GPS data from 20 July to 10 August for determination orbit. Then, we analyzed the residual variation of the POD processing for the HY2D satellite, presented the overlap orbit precision and compared the calculated orbits with precise orbit products provided by CNES, and showed the SLR validation results.

### 4.1. POD Residuals Analysis

Because the change in daily residuals of POD is basically similar for the HY2D satellite, we present the residual variation of observations for two days of data calculated by the above method and model strategies. Additionally, the selected days are, respectively, July 10 and 21, 2021 for BDS and GPS. Figure 4 shows the residual variation in spaceborne BDS phase observations from HY2D satellite on 10 July 2021. As can be seen from the figure, about 96.41 percent of residual values are located within ±15.00 mm. The average value and standard deviation of residuals are 0.036 mm and 9.12 mm, respectively, which show that these residuals have no significant deviations.

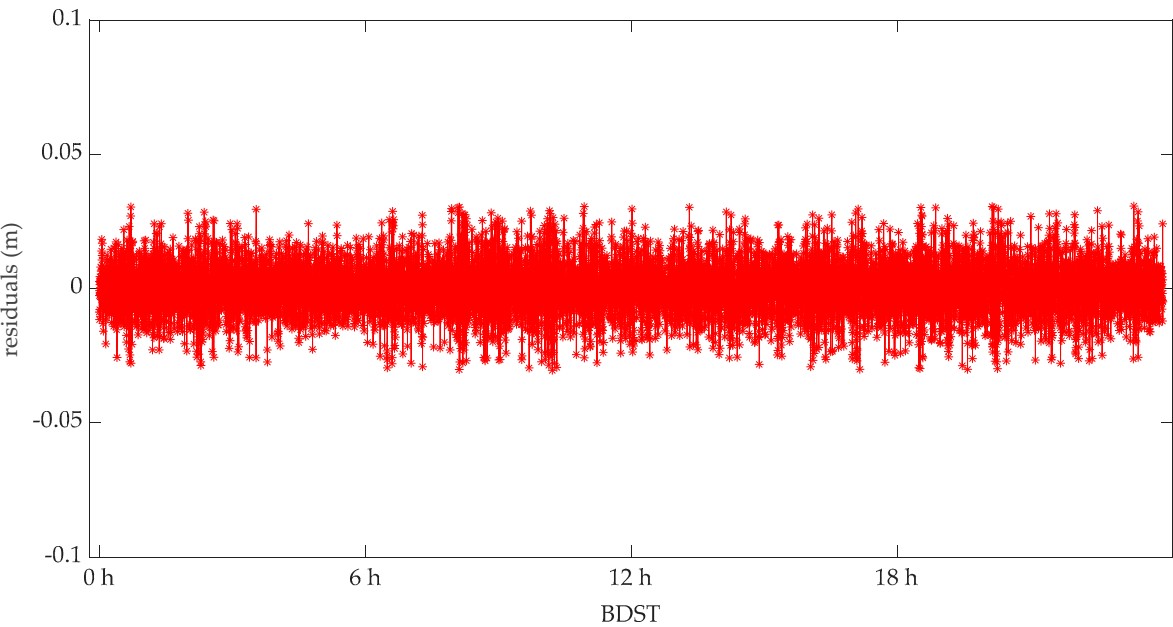

**Figure 4.** Variation of residuals in the POD for HY2D using spaceborne BDS data.

Moreover, Figure 5 shows the residuals variation in spaceborne GPS phase observations from the HY2D satellite on 21 July 2021. As can be seen from the figure, about 97.21 percent of residuals values are located within ±15.00 mm, similar to BDS. After the statistical calculation of these residuals, their average value and standard deviation are 0.026 mm and 8.53 mm, respectively, and there are also no obvious systematic errors in the residuals of GPS.

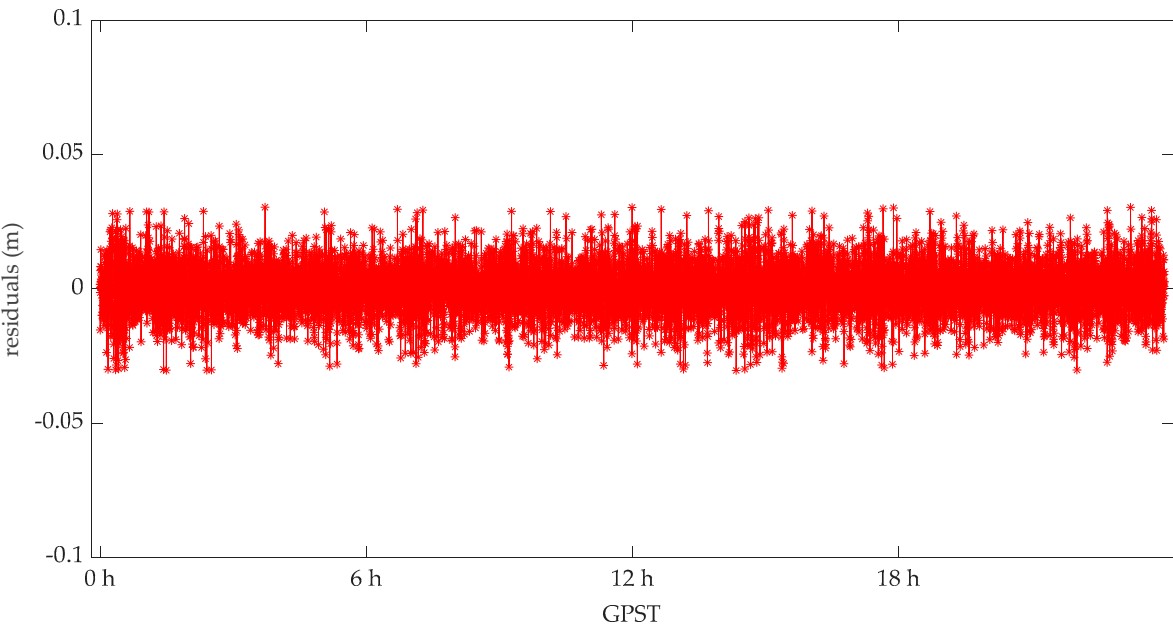

**Figure 5.** Variation of residuals in the POD for HY2D using spaceborne GPS data.

### 4.2. POD Precision Analysis for HY2D

In this study, we also compare the calculated orbit using spaceborne BDS and GPS data with the precise Doppler Orbitography and Radio-positioning Integrated by Satellite (DORIS)-derived orbits provided by CNES. Figures 6 and 7, respectively, show the position differences in Radial (R), Tangential (T), and Normal (N) directions between the calculated orbit and the precise orbit.

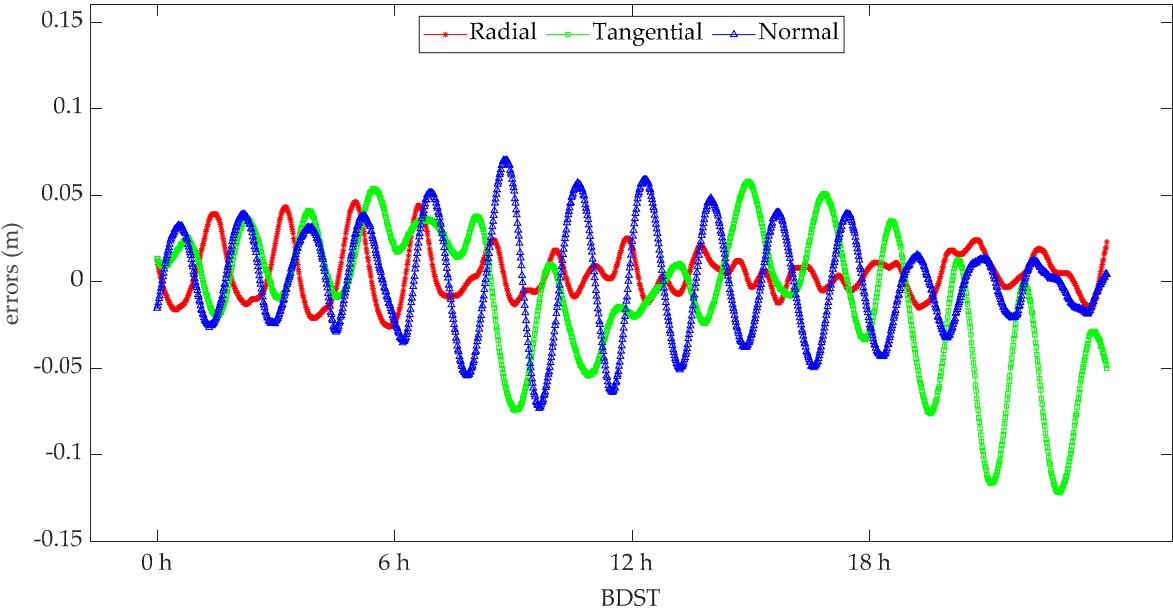

**Figure 6.** The variation in errors for the HY2D satellite using spaceborne BDS data compared with CNES orbit.

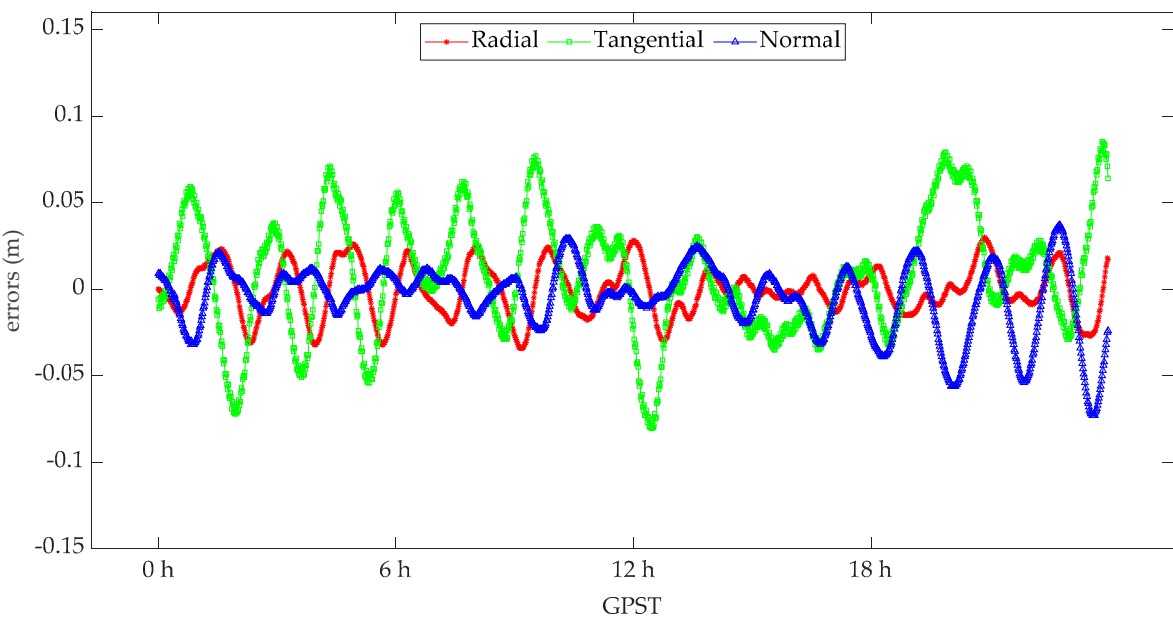

**Figure 7.** The variation in errors for the HY2D satellite using spaceborne GPS data compared with CNES orbit.

As can be seen from Figures 6 and 7, the variations in the radial direction have smaller fluctuation, which are basically located within ±0.03 m. However, the fluctuation in the tangential direction is relatively bigger, and this may be related to the used models of atmospheric drag and solar pressure, which are hard to estimate exactly. It should be noted that the normal fluctuations of GPS are relatively smaller than BDS; this is related to the number of GPS satellites, which are more than the BDS satellites and the orbit precision of GEO and IGSO satellites is lower. To analyze the precision visually, Table 5 gives the accuracy statistics of the HY2D satellite using spaceborne BDS and GPS data. We can find that the mean values are close to zero, which illustrate that these error values are basically unbiased and have no obvious systematic errors. For spaceborne BDS data, the radial, tangential, normal and 3D accuracy can achieve 1.5 cm, 4.1 cm, 3.0 cm, and 5.3 cm, respectively, and the radial, tangential, normal and 3D accuracy are, respectively 1.5 cm, 3.5 cm, 2.0 cm, and 4.3 cm for spaceborne GPS data. However, because of the fewer satellites available, the consistency the BDS-derived orbits are slightly worse than the GPS-derived orbits.

**Table 5.** Accuracy statistics of POD for HY2D using spaceborne BDS and GPS data (unit: cm).

|  |  | Mean | RMS |
|---|---|---|---|
| BDS | Radial | 0.4 | 1.5 |
|  | Tangential | −0.8 | 4.1 |
|  | Normal | 0.1 | 3.0 |
|  | 3DRMS | 5.3 |  |
| GPS | Radial | −0.1 | 1.5 |
|  | Tangential | 0.8 | 3.5 |
|  | Normal | −0.5 | 2.0 |
|  | 3DRMS | 4.3 |  |

For validating the reliability of the used method and model strategies, we also utilize more spaceborne BDS and GPS data to conduct POD solutions for HY2D satellite, and the results of comparison with precise orbit provided by CNES are shown in Figures 8 and 9 in radial, tangential, normal and 3D directions.

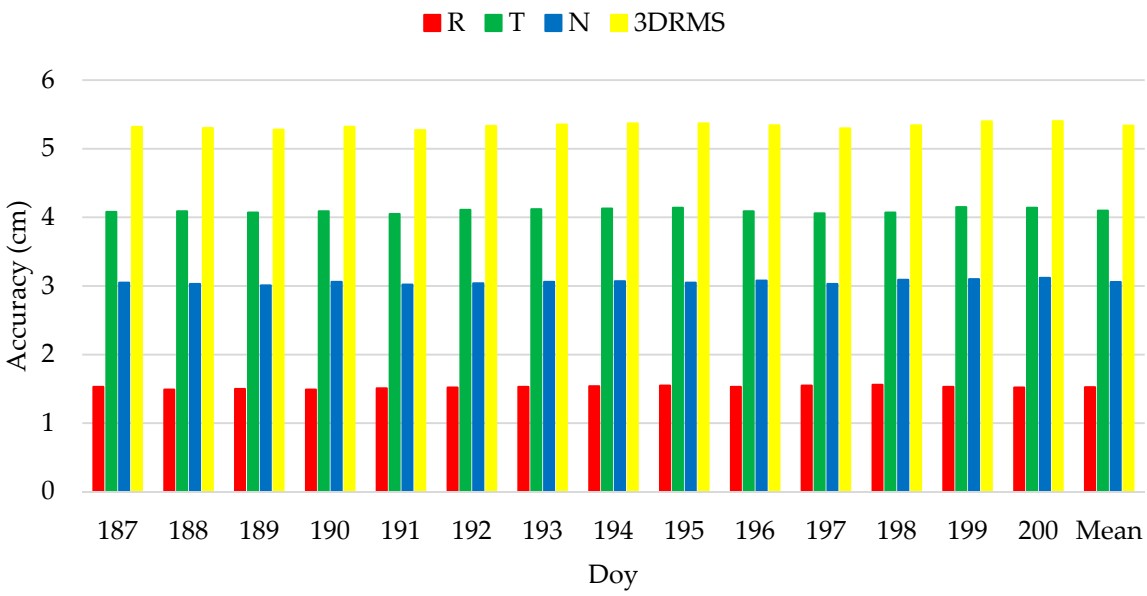

**Figure 8.** The RMS values for HY2D satellite using spaceborne BDS data.

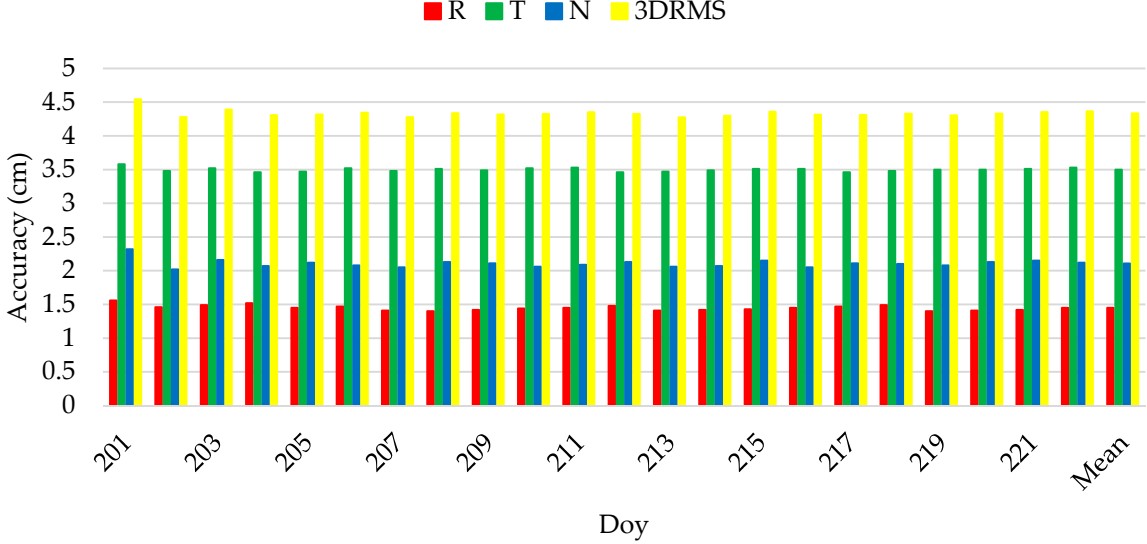

**Figure 9.** The RMS values for HY2D satellite using spaceborne GPS data.

As can be shown from Figures 8 and 9, the radial RMS values are basically 1.4~1.5 cm for BDS and GPS data, and the tangential and normal accuracies are slightly poorer compared to the radial direction. Overall, the precision based on spaceborne GPS data are slightly better than that of spaceborne BDS data, and the results are in agreement with the previous analysis, which shows that the used method and model strategies have a certain reliability. Moreover, Tables 6 and 7 show the accuracy statistics of HY2D. As can be shown from Tables 6 and 7, the average radial accuracy based on spaceborne BDS and GPS data are, respectively, 1.5 cm and 1.4 cm, the average three-dimensional accuracy are 5.3 cm and 4.3 cm, respectively. These results illustrate that the HY2D satellite, using spaceborne BDS and GPS data, can achieve the radial precision of 1.4~1.5 cm, the 3D position precision better than 5.5 cm, and the radial precision can satisfy high-precision altimetry applications.

**Table 6.** The precision statistics of HY2D satellite using spaceborne BDS data (unit: cm).

| Doy | R | T | N | 3DRMS |
|---|---|---|---|---|
| 187 | 1.5 | 4.1 | 3.1 | 5.3 |
| 188 | 1.5 | 4.1 | 3.0 | 5.3 |
| 189 | 1.5 | 4.1 | 3.0 | 5.3 |
| 190 | 1.5 | 4.1 | 3.1 | 5.3 |
| 191 | 1.5 | 4.1 | 3.0 | 5.3 |
| 192 | 1.5 | 4.1 | 3.0 | 5.3 |
| 193 | 1.5 | 4.1 | 3.1 | 5.4 |
| 194 | 1.5 | 4.1 | 3.1 | 5.4 |
| 195 | 1.6 | 4.1 | 3.1 | 5.4 |
| 196 | 1.5 | 4.1 | 3.1 | 5.3 |
| 197 | 1.5 | 4.1 | 3.1 | 5.3 |
| 198 | 1.6 | 4.1 | 3.1 | 5.3 |
| 199 | 1.5 | 4.2 | 3.1 | 5.4 |
| 200 | 1.5 | 4.1 | 3.1 | 5.4 |
| Mean | 1.5 | 4.1 | 3.1 | 5.4 |

**Table 7.** The statistics of accuracy for HY2D satellite using spaceborne GPS data (unit: cm).

| Doy | R | T | N | 3DRMS |
|---|---|---|---|---|
| 201 | 1.6 | 3.6 | 2.3 | 4.5 |
| 202 | 1.5 | 3.5 | 2.0 | 4.3 |
| 203 | 1.5 | 3.5 | 2.2 | 4.4 |
| 204 | 1.5 | 3.5 | 2.1 | 4.3 |
| 205 | 1.5 | 3.5 | 2.1 | 4.3 |
| 206 | 1.5 | 3.5 | 2.1 | 4.3 |
| 207 | 1.4 | 3.5 | 2.1 | 4.3 |
| 208 | 1.4 | 3.5 | 2.1 | 4.3 |
| 209 | 1.4 | 3.5 | 2.1 | 4.3 |
| 210 | 1.4 | 3.5 | 2.1 | 4.3 |
| 211 | 1.5 | 3.5 | 2.1 | 4.4 |
| 212 | 1.5 | 3.5 | 2.1 | 4.3 |
| 213 | 1.4 | 3.5 | 2.1 | 4.3 |
| 214 | 1.4 | 3.5 | 2.1 | 4.3 |
| 215 | 1.4 | 3.5 | 2.2 | 4.4 |
| 216 | 1.5 | 3.5 | 2.1 | 4.3 |
| 217 | 1.5 | 3.5 | 2.1 | 4.3 |
| 218 | 1.5 | 3.5 | 2.1 | 4.3 |
| 219 | 1.4 | 3.5 | 2.1 | 4.3 |
| 220 | 1.4 | 3.5 | 2.1 | 4.3 |
| 221 | 1.4 | 3.5 | 2.1 | 4.4 |
| 222 | 1.5 | 3.5 | 2.1 | 4.4 |
| Mean | 1.4 | 3.5 | 2.1 | 4.3 |

*4.3. SLR Validation for the POD of HY2D*

SLR is an independent measurement technique compared to GNSS, and thus SLR validation can validate objectively the accuracy and reliability of GNSS-derived orbits [11,38]. Here, we select the SLR normal point data from the corresponding periods of the HY2D satellite (download: ftp://edc.dgfi.tum.de/pub/slr/data/npt_crd/, accessed on 10 February 2022). The SLR residual results cannot verify the accuracies of the components in each direction or orbital position directly, but they denote that the distance precision between the given SLR station and the validated satellite. When the higher cut-off angle is set, the SLR residuals can better reflect the radial accuracy. Because HY2D is an altimetry satellite, its radial precision is our primary concern. In this study, we set the cut-off angles as 20, and give the time series of SLR residuals in Figure 10.

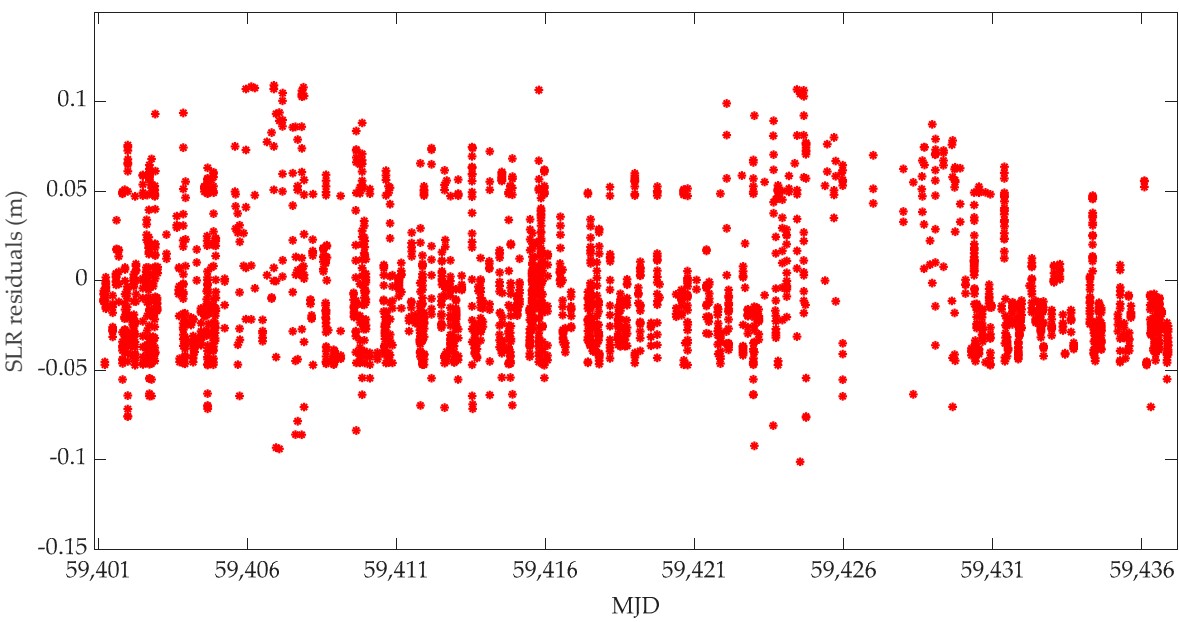

**Figure 10.** The variation of SLR residuals for HY2D satellite.

As can be seen from Figure 10, most of the residual values are located within ±8.0 cm, and have no obvious abnormal value. The orbits of the first 14 days are calculated using spaceborne BDS data, and the next 22 days orbits are calculated using spaceborne GPS data. The range of SLR residuals for BDS-derived orbits is slightly bigger than that of GPS-derived orbits, which is related to that the precision based on spaceborne GPS data are slightly better than that of spaceborne BDS data. After the statistical calculation of these residuals, their average value and standard deviation are, respectively, −0.7 cm and 3.3 cm, which illustrate that these residuals have no significant deviations. The results also indicate that the orbital accuracy of HY2D satellite can satisfy the current high-precision altimetry requirements.

## 5. Conclusions

Based on the spaceborne BDS and GPS data of the newly launched HY2D satellite, this paper conducts research into POD using one month of data. The model strategies are developed to solve the precise orbits of the HY2D satellite, and this study selects a month of real onboard GPS and BDS data to validate the developed model strategies. The main conclusions of the study are given in the following:

(1)  The standard deviation of the POD residuals based on spaceborne BDS and GPS data are, respectively, 9.12 mm and 8.53 mm, and these residual variations show no significant deviations or systematic errors.

(2)  The comparison results with DORIS-derived orbits show that the average radial RMS value of spaceborne BDS and GPS are 1.5 cm and 1.4 cm, respectively, and the corresponding 3D RMS accuracy are 5.3 cm and 4.3 cm, respectively. Overall, these results indicate that the POD processing of the HY2D satellite using spaceborne BDS and GPS data can achieve 1 cm radial precision and satisfy the current high-precision altimetry applications.

(3)  According to the SLR validation results, it is shown that the standard deviation of residuals is 3.3 cm, which indicates that the orbital accuracy of the HY2D satellite is approximately 3.3 cm, and the used model strategy is also reliable.

As an example of the HY2D satellite, this study adopts spaceborne BDS and GPS data to conduct POD research. Eventually, the model strategies are developed to solve the precise orbit of the HY2D satellite. Meanwhile, we use the external DORIS-derived precise

orbit of the HY2D satellite and SLR normal point data to validate the calculated orbit, and the results show that the HY2D satellite can achieve 1 cm radial precision.

**Author Contributions:** C.Z. provided the initial idea for this research; H.P., C.Z., S.Z. and X.Z. collected the experimental data and conducted the experiment; C.Z., H.P., S.Z., B.P., H.Y. analyzed the results of the experiment; C.Z., J.Z., J.H., F.G. and R.C. wrote the paper. All authors have read and agreed to the published version of the manuscript.

**Funding:** This work was supported by the National Key Research Program of China "Collaborative Precision Positioning Project" (No. 2016YFB0501900), the National Natural Science Foundation of China (Grant No. 42174222, 41904165, 62101219, 41804019), the State Key Laboratory of Geodesy and Earth's Dynamics self-deployment project (No. S21L6101, S21L8101), the Natural Science Foundation of Hubei Province (No. 2017CFB372), the Natural Science Foundation of Jiangsu Province (No. BK20210921), the Natural Science Foundation of Fujian Province (No. 2018J01480).

**Institutional Review Board Statement:** Not applicable.

**Informed Consent Statement:** Not applicable.

**Data Availability Statement:** Not applicable.

**Acknowledgments:** The authors acknowledge the satellite information of HY2D provide by NSOAS. Our sincere thanks to the NSOAS for providing space-borne GPS and BDS data for HY2D; CODE for providing GPS satellite orbits, clocks and Earth rotation parameters; the CNES for providing precise orbits for HY2D; the ILRS for providing SLR data of the HY2D satellite. Meanwhile, we would like to thank the anonymous reviewers for their valuable comments.

**Conflicts of Interest:** The authors declare that they have no known competing financial interests or personal relationships that could have appeared to influence the work reported in this paper.

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
