# Peer review of "Analysis of Precise Orbit Determination for the HY2D Satellite Using Onboard GPS/BDS Observations"

_remotesensing, doi:10.3390/rs14061390_

Round 1

Reviewer 1 Report

See the attached PDF file.

Author Response

Dear Reviewer,

We appreciate to resubmit our manuscript entitled “Analysis of Precise Orbit Determination for the HY2D Satellite Using Onboard GPS/BDS Observations”.

Thank you very much for your comments about the manuscript. The authors have modified the article according to your comments. All changes have been marked in the article.

We deeply appreciate your consideration of our manuscript. If you have any queries, please contact me at the address below. Thank you very much for your attention. We are looking forward for your response sincerely.

Best regards.

Yours sincerely

Chongchong Zhou

Reviewer 2 Report

The authors want to determine precise orbit data for the chinese LEO satellite HY2D. Based on theory and experimental data they report very good results, therefore the paper deserves publication.

However, as is now, I find difficult to read it because of several issues, which they can easily correct. In Section 1, the purpose of the paper is not very much highlighted. It is found just at the end, in a single sentence.

The notes reported at the foot of the Tables are their captions, therefore they must be written at the top of the tables.

A list of abbreviations with their "decoding" must be reported, e.g. in an Appendix.

Table 7 is trivial, shows only two numbers!

Can the authors discuss how much their accurate results affect  the six classical orbital elements?

In the review of Section 1, they should mention the possible use of the so-called "GeoSurf" constellation, published in Future Internet, January 2020.

Author Response

(The authors gave the same response as above.)

Reviewer 3 Report

Manuscript ID: remotesensing-1618668

Title: Analysis of Precise Orbit Determination for the HY2D Satellite Using Onboard GPS/BDS Observations

Authors: Hailong Peng, Chongchong Zhou, Shiming Zhong, Bibo Peng, Xuhua Zhou, Haoming Yan, Jie Zhang, Jinyang Han, Fengcheng Guo, and Runjing Chen

In this study authors use one month of spaceborne data of altimetry satellite HY2D of China to perform precise orbit determination (POD) process at high level accuracy.

In the present form this manuscript cannot be accepted for publication on Remote Sensing, I therefore suggest some modifications and fixes:

As general remarks:

  • I suggest a revision of the English language by a native English speaker, authors can refer to the journal service
  • This paper is structured as a technical note more than an article.
  • The accuracy of the methods presented here are at the level of the centimetre or better but no more than one mm, I therefore suggest authors to indicate all the number in this paper using centimetres with no more than one decimal place (ex: 3.5 cm, see line 81 of the Introduction).

In particular:

  • At line 47 of the Introduction – I suggest authors to use a different verb than master like obtain, detect and so on.
  • At line 72 – numbers are expressed in cm and with two decimal places meanwhile at line 81 the number 3.5 cm is indicated. I therefore suggest authors to uniform all the numbers in the body if this manuscript using the centimetres and only one decimal place as in line 81 of the Introduction)
  • At line 103 of the Methods, the reference [23] doesn’t exist, I think that authors should refer to the following link: https://ilrs.gsfc.nasa.gov/missions/satellite_missions/current_missions/hy2d_general.html

  • At line 121 - Table 4 – I suggest authors to convert the table data in centimetres (see point (2) and general remarks).

  • At line 123 of the Table 4 caption, I think that the word “gives” should be changed in “given”.

  • In the sentence from line 127 to line 130, I suggest authors to add some references.

  • At line 136 – Table 2 – I suggest authors to remove the third decimal place.

  • At line 150 – Authors should convert the data of Figure 2 in a Table.

  • At lines 163 and 167 – Authors should improve the quality of the Figures 3 and 4 (axis label format, fonts, symbols etc. etc.).

  • At line 224 – I think that the usage of verb “pointed out” is better than “pointed”.

  • In the section “Discussion and Conclusion” and in the Table 4, authors should convert all the numbers in centimetres using only one decimal place as indicated before.

  • At line 315 – I think that in the sentence “slightly better than the that of spaceborne BDS data” is better to eliminate the word “the”.

  • At lines 326 and 330, Tables 5 and 6, data should be expressed with only one decimal place (ex. 3.5 cm, see line 81).

  • At line 336, presently at the link indicated by authors (ftp server) only data of satellite HY-2A can be found, authors must solve/explain this discrepancy.

  • At line 343 – The corresponding Figure 11 can be improved.

  • At line 355 – The data in Table 7 can be expressed with only one decimal place.

  • At line 365 – In the section Conclusions, the same indication of the point 16).

Author Response

(The authors gave the same response as above.)

Round 2

Reviewer 1 Report

I am satisfied with the revision.

Author Response

(The authors gave the same response as above.)

Reviewer 3 Report

Manuscript ID: remotesensing-1618668

Title: Analysis of Precise Orbit Determination for the HY2D Satellite Using Onboard GPS/BDS Observations

Authors: Hailong Peng, Chongchong Zhou, Shiming Zhong, Bibo Peng, Xuhua Zhou, Haoming Yan, Jie Zhang, Jinyang Han, Fengcheng Guo, and Runjing Chen

In this study authors use one month of spaceborne data of altimetry satellite HY2D of China to perform precise orbit determination (POD) process at high level accuracy.

In the present form this manuscript cannot be accepted for publication on Remote Sensing, I therefore suggest some modifications and fixes:

As general remarks:

  • I strongly suggest a revision of the English Language by a native English speaker, authors can refer to the journal service.

  • In the present form this manuscript is still structured as a technical note more than an article, in this form cannot be accepted for publication on Remote Sensing.

  • The accuracy of the methods presented here, in my opinion, are at the level of the centimetre or better but no more than one mm, I therefore suggest authors to indicate all the numbers in this manuscript (also in the Tables and Figures, everywhere) using centimetres with only one decimal place.

  • In the introduction it is necessary to add some sentences with references to clarify the real accuracy of the methods presented in this work, tents of millimetres, millimetres or centimetres?

As specific indications:

  • From line 150 to 153 – Authors should indicate the described data adding a Table.

  • At lines 160 and 166 – Authors must improve the quality of the Figures 2 and 3 (axis labels, format, fonts, symbols, etc. etc.) .

  • At lines 203 to 205 authors should verify the format of the equation (2), in fact there are some characters confused/unreadable.

  • At line 345 – The corresponding Figure 10 must be improved (axis labels, format, fonts, symbols) as for Figures 2 and 3.

Author Response

Dear Reviewer,

We appreciate to resubmit our manuscript entitled “Analysis of Precise Orbit Determination for the HY2D Satellite Using Onboard GPS/BDS Observations”.

Thank you very much for your comments about the manuscript. The authors have modified the article according to your comments. All changes have been marked in the article.

We deeply appreciate your consideration of our manuscript. If you have any queries, please contact me at the address below. Thank you very much for your attention. We are looking forward for your response sincerely.

Best regards.

Yours sincerely

Chongchong Zhou

This manuscript is a resubmission of an earlier submission. The following is a list of the peer review reports and author responses from that submission.